# Reinforcement Corrosion in RC Hollow Piers: Destructive and Non-Destructive Tests

**DOI:** 10.3390/ma16072790

**Published:** 2023-03-31

**Authors:** Gian Piero Lignola, Francesco Fabbrocino, Andrea Prota, Edoardo Cosenza, Gaetano Manfredi

**Affiliations:** 1Department of Structures for Engineering and Architecture, University of Naples “Federico II”, 80125 Napoli, Italy; 2Department of Engineering, Telematic University Pegaso, 80143 Napoli, Italy

**Keywords:** corrosion, experimental test, hollow bridge piers, monitoring, structural safety, theoretical model

## Abstract

In this work, cyclic-load tests on reduced-scale corroded reinforced-concrete hollow cross-section bridge piers have been experimentally performed and compared to the results of similar non-corroded piers. Piers were aged by using an imposed electric current and sodium chloride water solution before performing a mechanical cyclic-load test. The corrosion process has been detected with Non-Destructive Evaluation techniques by means of SonReb method (to check concrete degradation) and by measuring corrosion potential (to check steel degradation). The crack pattern was recorded by dedicated cameras, and an LVDT system was set up to monitor the cyclic-load test. Experimental results focused on degradation monitoring and mechanical performance under cyclic loads. During the cyclic-load mechanical test, the first cracks on the piers surface occurred diagonally, inclined at about 45°. This is the consequence of the failure mode change from ductile failure, as expected for slender designed piers, to brittle shear failure. The flexural failure occurred in the case of non-corroded piers. Presented tests can provide a useful contribution of experimental data to analyse the behaviour of corroded reinforced concrete hollow bridge piers, scarcely tested. In particular, the cyclic response can be a useful reference for the proposition/validation of nonlinear capacity models for the evaluation of the seismic capacity of corroded bridge piers.

## 1. Introduction

Infrastructures built before the 1980s were designed without taking into account the modern principles of seismic engineering and basic regulations or design codes to counteract seismic loads were available. Hence, the assessment of seismic performance shall be a priority for such infrastructures. The conservation and utilization of the infrastructure network after an earthquake is essential to prevent the interruption of the main rescue lines, given the difficulty for the arrival of relief and often the significant economic loss in the affected areas that are isolated until the reconstruction/restoration.

In Italy and in many other countries, a commonly used construction technique for piers is the hollow reinforced concrete (RC) pier that provides, at the same time, strength and reduced mass compared to solid sections. Seismic energy dissipation is assigned in the design of ductile hinges at the base [1,2]. The negative element for this type of structure is the considerable deformation due to shear, which also represents considerable percentage of the total displacement at the top of the pier [3]. Numerous experimental studies have addressed the problem of the effect on piers of cyclic forces from wind and vehicular traffic: the effect is a combined shear and bending effect or shear failures at the base for both circular and rectangular sections [4,5,6]. A wide variety of literature also exists for hollow sections, and, in addition, previous studies [7,8,9] have also detected several factors (namely, inner to outer cross-section dimensions, axial load ratio, longitudinal and transverse reinforcement ratios) that influence the seismic behaviour of the bridge piers, referring to the real geometries of the RC hollow structures.

Corrosion of the reinforcement was found to be the predominant deterioration mechanism for RC structures, impacting infrastructure resilience [10], and this has potentially critical consequences for safety and serviceability [11,12,13], with impacts on sectional strength [14,15,16] and ductility [17,18] capacities. The degradation of the materials (steel and/or concrete) decreases the overall seismic performance of bridges [19,20] and different corrosion levels, due to exposition time and environment [21], or uniformity of corrosion [22], or retrofitting with composite jackets [23] have a clear impact on failure mode and seismic capacity in general.

Few results are available for corrosion of RC bridge piers with hollow cross sections: in particular, the corrosion of steel rebars determined a significant reduction in the lateral strength, secant stiffness and energy loss per cycle of the piers, both experimentally [24,25] and numerically [25]. The effect of jacketing as corrosion protection, other than retrofitting [26], was analysed as well [27]. Corrosion of stirrups eliminates the confinement of the transverse reinforcement, leading to brittle failure mode changing from ductile failure, as expected for slender designed piers, to brittle shear failure [28], as it was already clear that the inelastic cyclic demand reduces the shear strength in RC solid piers [29].

In addition, Structural Health Monitoring (SHM), focusing on damage investigation with non-destructive evaluation techniques, has been adopted to monitor concrete degradation [30], reinforcement corrosion [31] and structures [32] as a crucial tool coupled with mechanical evaluations. Ultrasonic Pulse Velocity (UPV) and Rebound Hammer (RH) are combined in the so-called SonReb method by the researchers to estimate the concrete compressive strength according to most frequently adopted RILEM standard [33] and other authors’ formulations [34].

The reduced availability of experimental results on hollow-cross-section RC piers induced the authors to perform this experimental program. In this study, the analysis of the structural response of the hollow piers (both circular and rectangular) of bridges, under cyclic forces at the upper end and influenced by the aging process at the base of the pier, is performed both by means of non-destructive monitoring and mechanical test evaluations.

## 2. Materials and Methods

### 2.1. Sections Geometry

This study investigated two 1:3 scaled hollow piers after corrosion, similar to previously tested non-corroded counterparts [35]: one with circular section and the other one with rectangular section, identified with the acronyms P12 and P14, respectively. The rectangular hollow pier has an overall pier specimen height (sum of base, stem and load head) of 2.40 m (1.35 m stem only) with an external cross section of (0.40 × 0.60) m^2^ and an internal hole cross section of (0.20 × 0.40) m^2^. The longitudinal reinforcement consists of two layers: an external one consisting of 18D8 and an internal one consisting of 10D8, while the D3 stirrups were positioned at a distance of 12 cm (Figure 1). The concrete cover, preserving the scale factor of the pier, was 1.7 cm conforming to the ratio of the thickness of the concrete cover to the size of the pier [36].

The circular pier with a circular hollow section has an overall height (base, stem and load head) of 2.55 m (1.50 m stem only) with an external diameter of 0.55 m and an internal diameter of 0.35 m. The longitudinal reinforcement consists of 24D8 with D3 stirrups at 12 cm spacing and 1,7 cm concrete cover (Figure 2).

The load head dimensions are (60 × 80 × 30) cm^3^ for the rectangular pier and (75 × 75 × 30) cm^3^ for the circular pier.

In the investigation and to ensure a fixed constraint during the mechanical test after the aging process, an oversized base was considered with the following dimensions (120 × 120 × 75) cm^3^.

### 2.2. Materials Properties

Several casts were used for the construction of the piers. The base was made of concrete class C35/40 and ribbed bars class B450C; the stem and the load head made of concrete with 20 MPa average compressive cubic strength at 28 days, longitudinal ribbed bars (B450C), with average yield strength, f_ym_, of 505 MPa and smooth transverse bars with average yield strength, f_ym_, of 655 MPa.

### 2.3. Ageing of the Piers

In order to investigate the seismic response of corroded piers, specimens were subjected to an accelerated ageing process using 3.5% sodium chloride solution (in a tank built adjacent to the portion to be corroded, see Figure 2) [37] and imposition of an anodic electric current (of about 170 mA/m^2^ by means of potentiostat–galvanostat) such as to induce bar consumption of about 200 μm/year (1.00 mA/m^2^ induces 1.17 µm/year of corrosion penetration) [38,39] for eight months. Only part of the stem of the two piers was subjected to the ageing process. In particular, it affected only few bars up to a height of 50 cm from the base and, as shown in Figure 3, only the longitudinal bars circled in red were corroded. The bars have been completely electrically insulated, except in the part affected by the ageing process.

In order to assess the evolution of the ageing process, non-destructive tests were carried out using the SonReb method and measurements of the corrosion potential, both in areas close to the corroded portion (e.g., measuring points 1B and 3B for rectangular pier), and, for control purposes, closer to the head of the pier stem (e.g., measuring points 1A and 3A for rectangular pier) (Figure 4a).

### 2.4. Test Setup

The load tests were carried out through a hydraulic actuator (thrust capacity of 500 kN in compression and 300 kN in tension) under displacement control; horizontal displacement cycles under almost static conditions were applied along the centre line of the pier head. With a hinge constraint, the rotation between the actuator and the pier was allowed during the test.

Three D16 nominal high strength post-tensioned steel strands were fixed inside the foundation at the base.; on the other side the strands were loaded by means of a hydraulic jack (capacity of 500 kN) on the top of the pier to apply a constant vertical load to the pier during the cyclic horizontal loading. The vertical load was provided by a rigid steel plate on the pier head that allowed the vertical load to be distributed. The foundation was fixed to the strong laboratory floor with other 4 post-tensioned steel bars. The illustration of the test configuration is shown in Figure 5.

## 3. Results of Non-Destructive Tests

### 3.1. Rectangular Hollow Pier

During the period of the ageing process, the ultrasound velocities were detected and sclerometer tests were performed, three times, in five points named (1B-2B-3B-1A-3A), arranged according to the scheme of Figure 4a. The grid, shown in Figure 4a, corresponds to the position of the longitudinal bars (vertical lines) and the stirrups (horizontal lines spaced about 12 cm). These tests were carried out at time 0 (28 days after casting), 90 days after the start of the ageing process and 120 days after. Figure 6 shows the velocity curves of the direct transmission ultrasonic velocity vs. aging time, the slope of which is an indication of the degree of maturation and of the current degradation. The values measured in the vicinity of the saline solution (1B and 3B points) led to curves with steeper gradients than those far from the solution (1A and 3A above the solution and 2B at the solution level, but at about 40 cm from sodium chloride solution), indicating greater degradation. The curves show decreasing trends with the exception of points 1A and 3A which, after about 70 days, show stabilization of the ultrasonic velocity, probably related to concrete maturation. It is necessary to specify that ultrasonic measurements have always been carried out after a few days of power supply interruption and on dry surfaces, in order to minimize possible interferences. At the same time, in order to interpret the data with the SonReb method, sclerometric tests were also carried out. Finally, to test the characteristics of the reinforcement bars and their degradation over time, some potential mappings were performed (along the stem, about every 15 cm from the top of each corroded bar) and the results in mV vs. Cu/CuSO_4_ are shown in Figure 7. The reported colour scale ranges from green to red for safe values and high corrosion rate values, respectively. Each of the six rows corresponds to a mapping point from top to bottom, and each block A to E corresponds to a bar (according to Figure 3a), values in mV are reported for each of the measuring day (time origin is 28 days after casting). In particular, values above −200 mV vs. Cu/CuSO_4_ (negligible corrosion) are reported with green colour, values between −200 mV vs. Cu/CuSO_4_ and −350 mV vs. Cu/CuSO_4_ (corrosion) are reported with yellow colour, while values below −350 mV vs. Cu/CuSO_4_ (corrosion at high speed), are reported with red colour. After 26 days of testing, the recorded values are higher than −100 mV vs. Cu/CuSO_4_ for the entire length of the bars, indicating that the aggressive agents have not yet triggered the corrosion process. After 59 days of testing, the area close to the aggressive agents presents values that tend to yellow and red for all bars (even if close to the aggressive agents, only). The following measurements (90, 167, 198 and 241 test days) show that the degradation is progressively also affecting the reinforcement sections most likely to be impacted by aggressive agents, highlighting important criticalities in the sections closest to them, due to the high recorded potential values.

### 3.2. Circular Hollow Pier

The types of tests performed on the circular pier are the same as those for the rectangular hollow pier. The points where the SonReb tests were performed are four, two in the lower zone (1A and 2A) and two in the upper zone (3A and 4A). In particular, point 1A is placed in the area in direct contact with aggressive agents, point 2A is chosen at a distance of 30 cm from the aqueous solution, point 3A is placed at 20 cm below the loading head and point 4A at 55 cm from the head. These tests were carried out at time 0 (28 days after casting), 90 days and 120 days. In Figure 8, trends in ultrasonic velocity values are shown. As in the previous case, it can be seen that the ultrasound velocities are substantially stable far from the aggressive agents, while close to the aggressive solution, there is a sharp decrease, hence indicating degradation. The zero-time speed of the ultrasounds in position 2A is probably higher due to lower porosity of the concrete in that position. As in the previous case, tests for corrosion potential in mV vs. Cu/CuSO_4_ were carried out (Figure 9), for corroded bars A to G (see Figure 3b and Figure 4b) mapped at seven heights (each row from top to bottom of the stem). In this case too, it can be seen that, as the days of aging pass, the corrosion process occurs in the areas close to the aggressive solution and develops towards the more distant ones.

## 4. Cyclic Mechanical Test

Destructive tests were performed by means of a cyclic load at the head of the piers to evaluate the overall behaviour of the piers, after the aging process.

### The Cyclic Test: Test Protocol and Instrumentation

The cyclic test was carried out under displacement control. The test was divided as follows: for the rectangular hollow pier, the designed test protocol indicated in Table 1 was followed; for the circular one, the designed test protocol indicated in Table 2 has been performed. For both piers, the level of drift (ratio of displacement to shear span), displacement, number of performed cycles and displacement speed are reported, along with the duration of each cycle and the total duration of the test. To simulate the load of the upper structure, a constant axial load (5% of the axial capacity) was applied during the test.

A wire potentiometer system was used for displacement monitoring. The used system allowed for the monitoring of the actual displacement to be recorded at the head of the pier. LVDTs (linear potentiometers) were used, positioned in diagonal, in order to estimate the contribution of shear to the displacement, and others in order to control out-of-plane displacement of the pier (Figure 10).

## 5. Results of Destructive Tests

The force-displacement diagram was constructed using average force data (recorded by the load cell at the actuator) and displacements at the top of the pier.

Figure 11 shows the force-displacement diagram of the entire rectangular test; for each of the three cycles, the maximum force load peaks have been identified. Figure 12 shows the overlap of the maximum envelopes for circular pier test. The force-drift histogram was constructed taking into account the data acquired considering a shear span equal to the height of the pier, hence 1.50 or 1.65 m, to be compared with the displacement of the head. Figure 13 shows the force-drift diagram and Table 3 the relative values, relative to the rectangular pier. Between consecutive test cycles at the same drift, there is percentage loss of capacity for the piers.

For the determination of the overall behaviour of these elements, the well-known secant stiffness modulus, Es, has been taken into account as the slope of the line passing through the two peak points, negative and positive, of the same load-unloading cycle, calculated with Equation (1):Es = (F*_max_* − F*_min_*)/(Δ*_max_* − Δ*_min_*)(1)
in Equation (1) F*_max_* is the positive force at Δ*_max_* (maximum displacement), F*_min_* is the negative force at Δ*_min_* (minimum displacement).

Figure 14 shows the stiffness modulus diagram for the hollow rectangular pier, while Table 4 shows the percentage drop of the load bearing capacity of the circular hollow pier in relation to the cycles and drifts.

In next moment-curvature diagrams, the curvature, *χ*, is calculated with Equation (2):(2)$$\chi =\frac{\left(\frac{mean\left[\Delta_{min}^{R};\Delta_{min}^{L}\right]}{L_{gauge}}\right)-\left(\frac{mean\left[\Delta_{max}^{R};\Delta_{max}^{L}\right]}{L_{gauge}}\right)}{D}$$
in Equation (2) the Δ*_max_* and Δ*_min_*values recorded by the instrumentation positioned along different sides of the pier (right and left sides, with apex *R* and *L*, respectively), but placed at the same height, is averaged.

In addition to the symbols presented for Equation (1), *L_gauge_* is the distance between the measuring points; therefore, 600 mm, *D*, is the distance between two measuring points on the same side positioned at 0.06, 0.66 and 1.26 m, respectively, from the base of the hollow pier stem. Figure 15, Figure 16 and Figure 17 show moment-curvature diagrams, where moment is the top horizontal force multiplied by the shear span or lever arm (i.e., distance from the points at 0.06, 0.66 and 1.26 m, respectively, from the base).

For completeness of discussion, a cracking analysis was carried out by inspecting the images taken by a camera in front of the pier during the cyclic test. Through the identification of maximum displacement values, the related photographs were classified and correlated. Twelve peak points for positive and negative loads were identified for each of the loading-unloading cycles. The images taken during the first three drifts do not show visible damage to the piers and are considered, therefore, not significant; those taken afterwards (i.e., fourth drift and after) are shown. The state of the rectangular pier at the beginning of the test is shown in Figure 18a.

The first crack occurs at the fourth drift cycle, in a pre-peak phase at 145.66 kN and a displacement of 6.65 mm. Figure 19 shows the behaviour of the pier as a force-displacement diagram, from which an anomaly (circled in red) can be observed in the load path when the first crack is formed, Figure 18b. Figure 18c, conversely, shows the formation of cracks and the evolution up to the failure.

Figure 20 reports similar results for the circular pier.

At drift four, the first crack occurs, in a pre-peak phase at 83.30 kN and at a displacement of 12.80 mm. In addition, in Figure 21, the behaviour of the pier is shown by means of the force-displacement diagram, in which another anomaly (red circle) can be seen that reveals the formation of the first crack shown in Figure 20b. In Figure 20c, conversely, a photo is shown relative to next moments to clarify the crack pattern and the development up to the failure.

The analysis of the crack pattern shows a typical behaviour in coupled bending and shear condition. Both piers have been designed and manufactured with a reduced level of longitudinal reinforcement to have a ductile behaviour common to slender members, hence failing in bending, as occurred in the case of the non-corroded pier [35]. The effect of corrosion then gives rise to an interaction with the shear failure that is not observed in the similar non-corroded pier, resulting in a reduction in both load capacity and ductility.

## 6. Theoretical Modelling of Pier Capacity

The non-symmetric behaviour between push and pull side of the test is mainly due to the reduction in diameter (i.e., corrosion) of longitudinal bars. However, the flexural capacity is marginally reduced (about 5%) due to corrosion of longitudinal bars and cyclic effect; in fact, the reduction in the longitudinal bar diameter due to corrosion is about 6.5% (estimated from imposed electric current and related corrosion penetration) [38,39], and it is expected to be comparable to the flexural capacity reduction (in fact, the flexural capacity is almost proportional to tensile reinforcement area at such small axial load ratios).

On the contrary, the corrosion of the stirrups made of thin transverse reinforcement bars is the main reason of the failure mode, changing from desired flexure to brittle shear due to corrosion effect. To support such experimental outcome, the shear capacity evaluation is based on the Kowalsky and Priestley [40] model. According to the model, the shear capacity of RC piers is the sum of three components: contributions by concrete, transverse reinforcement and axial load, according to the Equation (3):(3)$$V_{n}=V_{c}+V_{s}+V_{p}.$$
each contribution corresponds to a different mechanism: *V_c_* accounts for shear carried by concrete, *V_s_* for the truss mechanism where shear is carried by the reinforcement and *V_p_* is the contribution provided by an inclined axial load strut, i.e., the compressed portion of the cross section can transmit the shear force at the basement. Each contribution in Equation (3) is provided by the following formulae (Equations (4)–(6)).
(4)$$V_{c}=\alpha \beta \gamma \sqrt{{f_{c}}^{\prime }}(0.8A_{g}),$$
(5)$$V_{s}=\frac{A_{v}f_{y}(D-c-\mathrm{cov})}{s}\cot\theta$$
(6)$$V_{p}=\frac{P(D-c)}{L_{v}}\;\mathrm{for}\;P\geq 0;V_{p}=0\;\mathrm{for}\;P<0,$$
where *f′_c_* is the concrete compressive strength, *f_y_* is the yield strength, *A_g_* is the gross area of concrete cross section, *A_v_* is the cross section of transverse reinforcement, *θ* is the cracking angle (suggested to be assumed as 30°), *D* is the section height, *c* is the neutral axis depth, cov is the concrete cover, s is the spacing of transversal reinforcement, *L_v_* is the shear span equal to pier height for a cantilever pier.

Terms *α* and *β* of the Equation (4), reported in Equations (7) and (8), account for aspect ratio and longitudinal steel ratio, respectively. Term *γ* in Equation (9) accounts for the degradation of the concrete shear resisting mechanisms with flexural ductility demand.
(7)$$1\leq \alpha =3-\frac{L_{v}}{D}\leq 1.5$$
(8)$$\beta =0.5+20\rho_{l}\leq 1$$
(9)γ=0.29−0.04(μ−2)

The non-corroded pier had ductile flexural failure, and it is confirmed by the theoretical model. In Figure 22, the shear capacity domain shows a reduction in shear capacity with flexural ductility demand; however, even at high ductility levels or drifts the shear capacity is higher than expected shear load. Conversely, the corroded pier suffered brittle shear failure after the flexural yielding at a drift level of 0.57%. In Figure 22, the shear capacity domain reduces also with the corrosion of stirrups, in fact, the reduction in the cross section of transverse reinforcement generates a reduction in shear contribution due to stirrups, *V_s_*. Based on experimental results, a shear crack opens clearly during the former seventh push cycle at a drift of 3.53% (see Figure 11), and this corresponds to a corrosion-induced reduction in stirrup cross-section of about 30%. In fact, the corrosion level of stirrups was estimated by a back analysis looking at the intersection of the first cycles envelope curve with the shear capacity domain corresponding to a reduction of 30% in the cross section of stirrups, mainly due to corrosion. It is worth noting that stirrups are external and more prone to corrosion than internal longitudinal bars.

The shear crack opening can be associated to imminent failure of the stirrups at the crack place. The force reversal reloaded the damaged stirrups (those crossing the crack under pushing force in the centre of the pier), but they did not contribute to the shear capacity. Therefore, the *V_s_* contribution to shear capacity is reduced not only by the corrosion effect, but by the additional loss of the damaged stirrups. A wide shear crack opens at the first pull cycle at a drift of −2.74% (see Figure 22) and corresponds, in the back analysis, to a reduction of about 50% in shear contribution of steel stirrups.

## 7. Conclusions

Experimental results of cyclic-load tests on reduced-scale corroded reinforced-concrete rectangular and circular hollow cross-section piers are analysed and compared to the results of similar non-corroded piers. The piers were corroded by using electric current and sodium chloride water solution before the mechanical cyclic-load tests. The corrosion process has been detected by performing the SonReb method (to check concrete degradation) and by measuring corrosion potential (to check steel degradation). Such non-destructive evaluation techniques revealed the degradation of concrete with a significant reduction in direct-transmission ultrasonic-wave velocity started after about 60 days from the beginning of the (accelerated) ageing, with clearly different trends close and far from the corroded portion. Similarly, corrosion potential of bars turns from safe values higher than −100 mV vs. Cu/CuSO_4_ for the entire length of the bars at the beginning of the process to values lower than −350 mV vs. Cu/CuSO_4_, revealing high corrosion rates, firstly, close to the aggressive agents and also far from them at the end of the process. These results confirm the predictability of corrosion degradation of steel and concrete by means of such techniques.

Corrosion induced a reduction of section in longitudinal and transverse bars, reducing both flexural and shear capacity. The transverse bars have lower section and concrete cover compared to longitudinal ones, hence are the most exposed steel reinforcement to corrosion with larger capacity losses.

After the cyclic-load test procedure, the first cracks on the piers’ surface occurred diagonally inclined at about 45°. This is the consequence of the failure mode change from ductile failure, as expected from slender designed piers with low longitudinal reinforcement ratio, to brittle shear failure, even with limited corrosion penetration. In fact, flexural failure occurred in the case of non-corroded piers. The global lateral capacity loss occurred both in terms of force and significant reduction in terms of ductility at low drift ratios.

The crack pattern was recorded by dedicated cameras and an LVDT system was set up to monitor the cyclic-load test. Experimental tests show that the behaviour of the piers, as expected, is almost similar for both kinds of cross-section.

Presented tests can provide a useful contribution to analyse the behaviour of corroded reinforced-concrete hollow bridge piers. In particular, the cyclic response can be a useful reference for the proposition/validation of nonlinear capacity models for the evaluation of the seismic capacity of corroded bridge piers.

## Figures and Tables

**Figure 1 materials-16-02790-f001:**
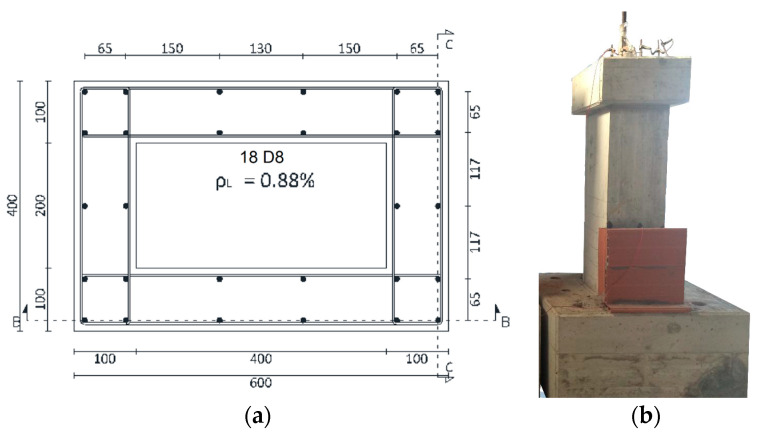
(**a**) Cross section (dimensions in mm) and (**b**) view of rectangular hollow pier with sodium chloride solution tank.

**Figure 2 materials-16-02790-f002:**
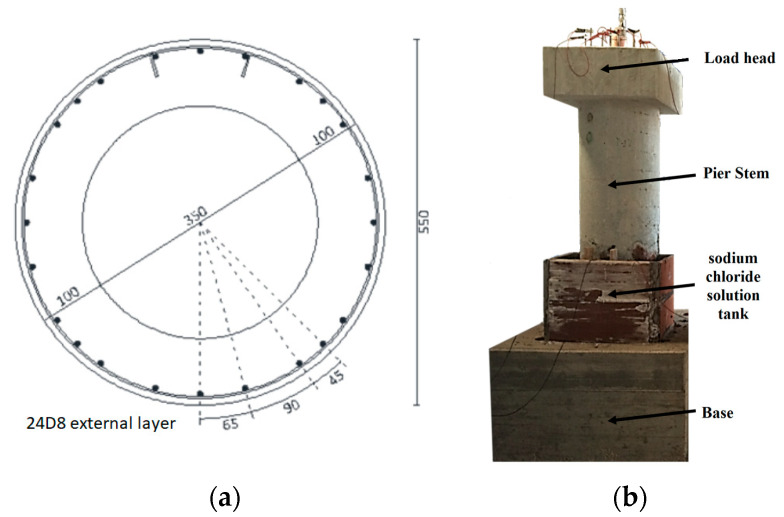
(**a**) Cross section (dimensions in mm) and (**b**) frontal view of circular hollow pier with sodium chloride solution tank.

**Figure 3 materials-16-02790-f003:**
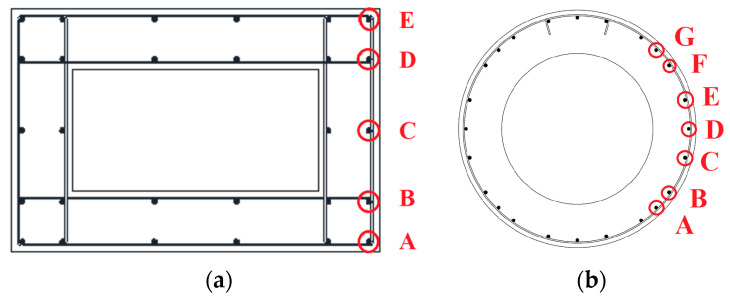
Corroded Bars (red circles): (**a**) rectangular and (**b**) circular hollow pier.

**Figure 4 materials-16-02790-f004:**
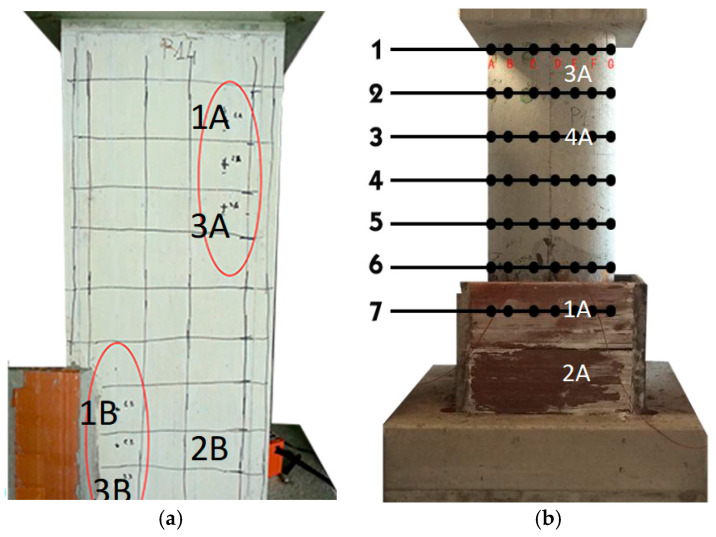
Measurements’ points for SonReb method and corrosion potential: (**a**) rectangular and (**b**) circular hollow pier.

**Figure 5 materials-16-02790-f005:**
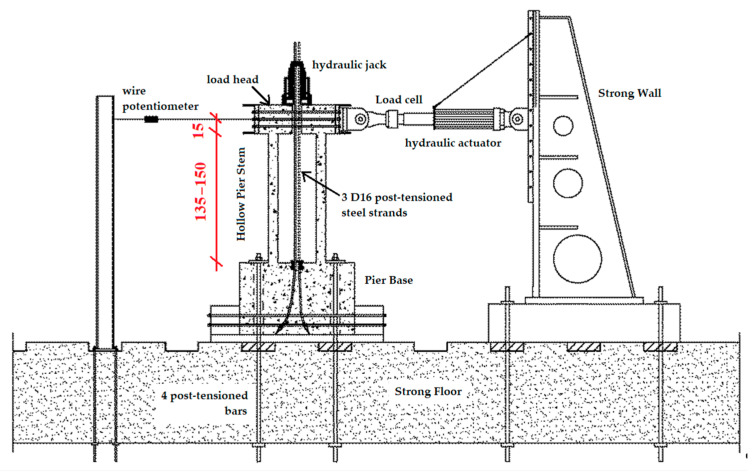
Cyclic load test setup scheme (dimensions in cm).

**Figure 6 materials-16-02790-f006:**
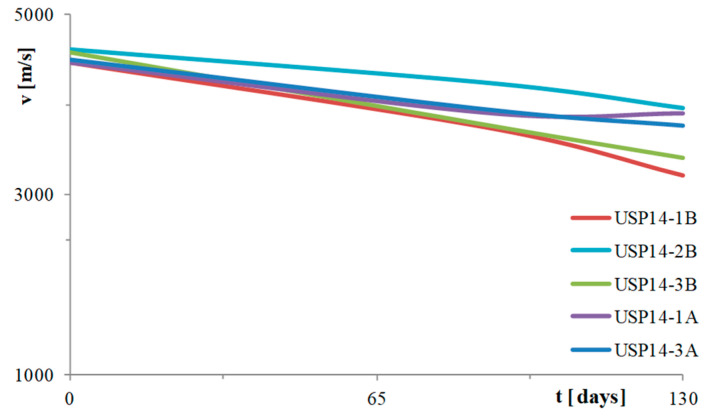
Ultrasound wave rate-time of ageing—rectangular pier.

**Figure 7 materials-16-02790-f007:**
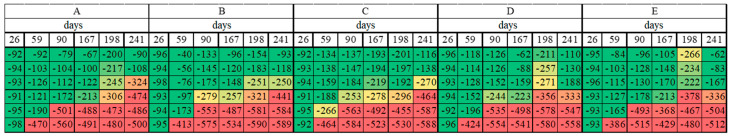
Corrosion potential of bars in mV vs. Cu/CuSO_4_—rectangular pier.

**Figure 8 materials-16-02790-f008:**
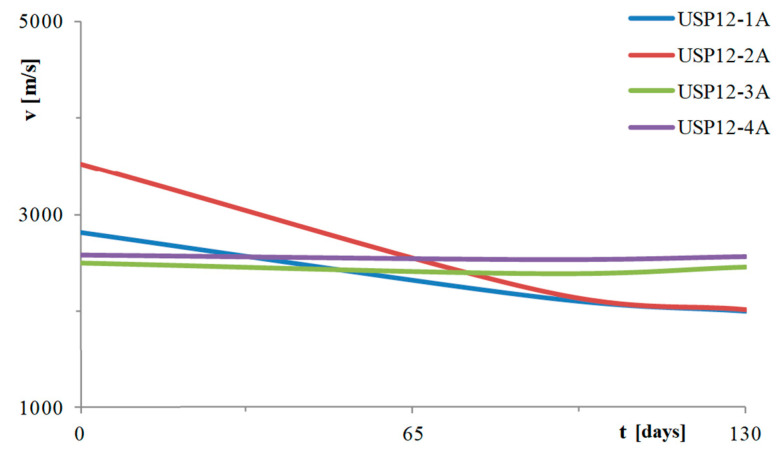
Ultrasound wave rate-time of ageing—circular pier.

**Figure 9 materials-16-02790-f009:**
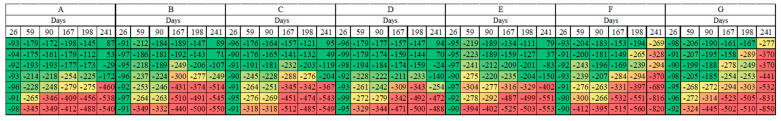
Corrosion Potential of bars in mV vs. Cu/CuSO_4_—circular pier.

**Figure 10 materials-16-02790-f010:**
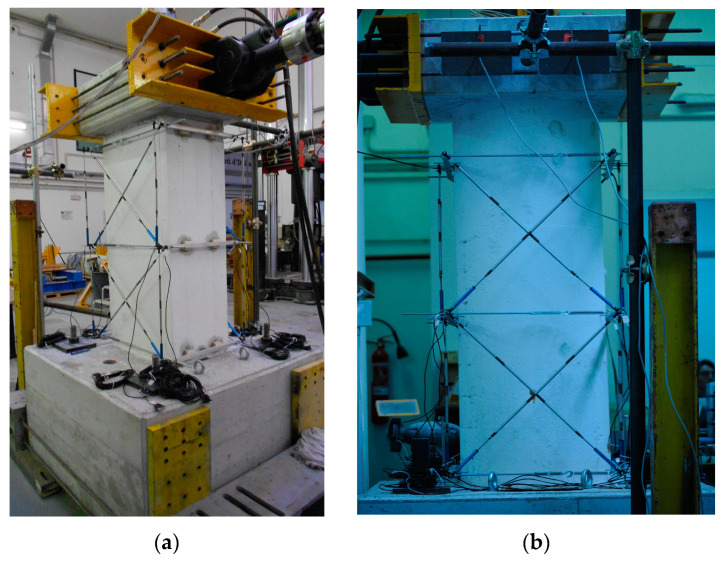
Monitoring system for cyclic test: (**a**) rectangular pier and (**b**) circular pier.

**Figure 11 materials-16-02790-f011:**
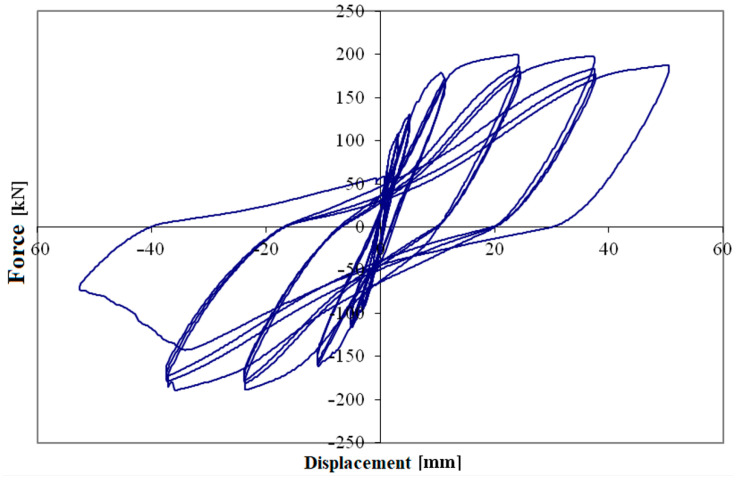
Load-displacement diagram of the rectangular pier test.

**Figure 12 materials-16-02790-f012:**
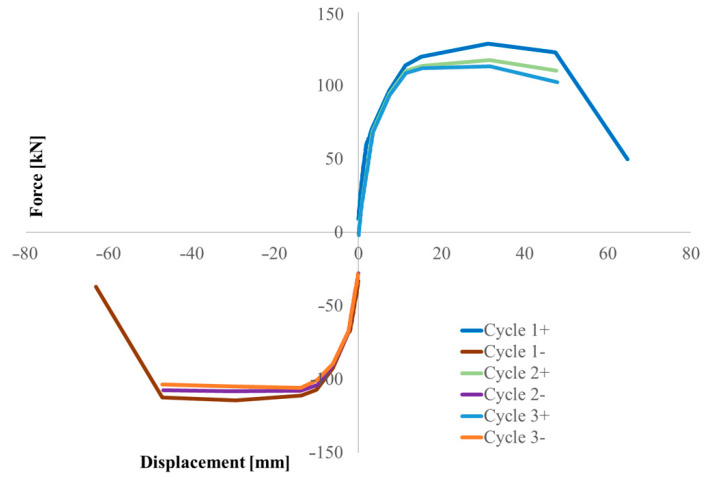
Horizontal force-peak displacements envelope diagram of the circular pier test.

**Figure 13 materials-16-02790-f013:**
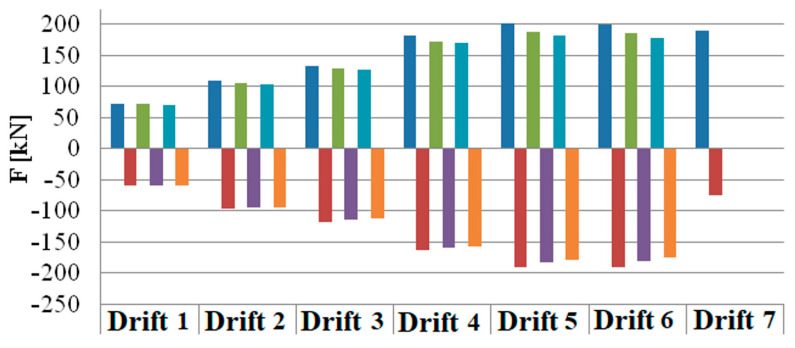
Load-drift diagram of the rectangular pier (for colour legend refer to Table 3).

**Figure 14 materials-16-02790-f014:**
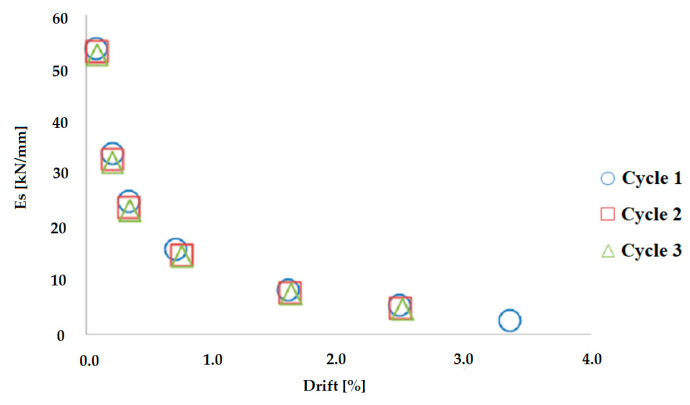
Secant modulus of elasticity-drift diagram—rectangular pier.

**Figure 15 materials-16-02790-f015:**
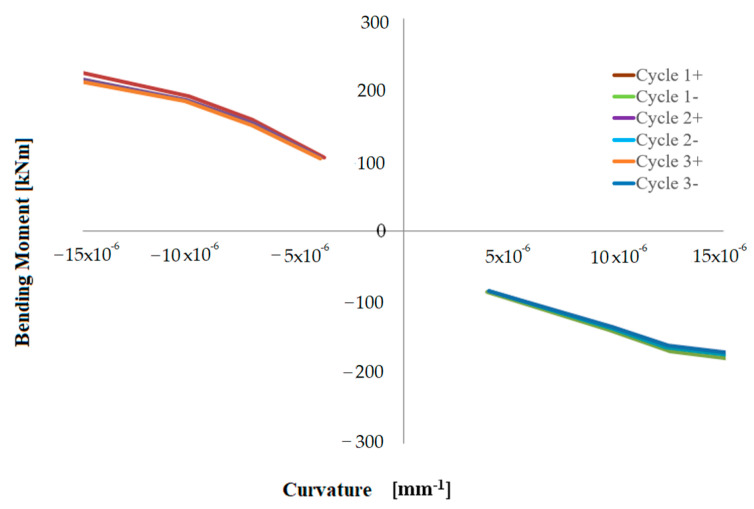
Bending moment-curvature diagram at the height of 0.06 m from the base—rectangular pier.

**Figure 16 materials-16-02790-f016:**
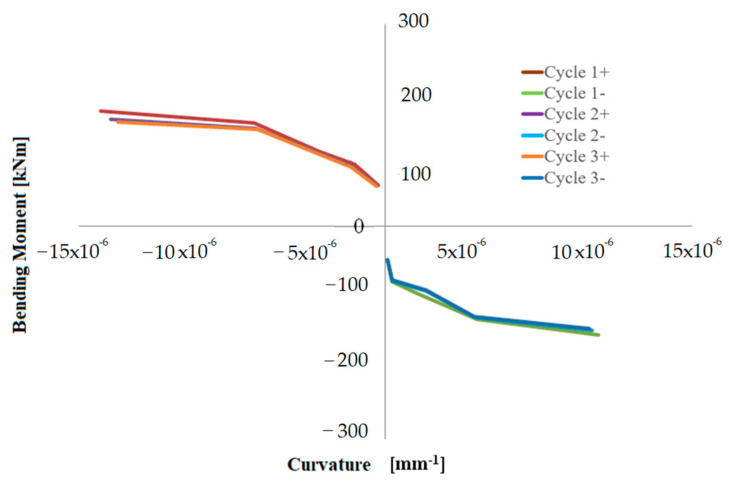
Bending moment-curvature diagram at the height of 0.66 m from the base—rectangular pier.

**Figure 17 materials-16-02790-f017:**
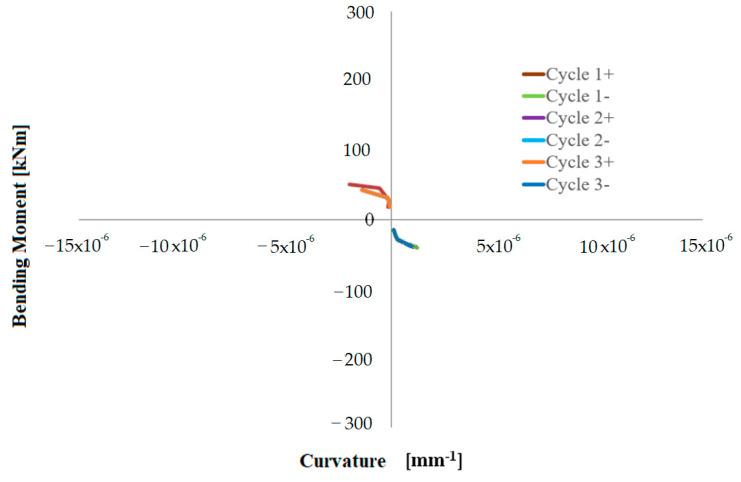
Bending moment-curvature diagram at the height of 1.26 m from the base– rectangular pier.

**Figure 18 materials-16-02790-f018:**
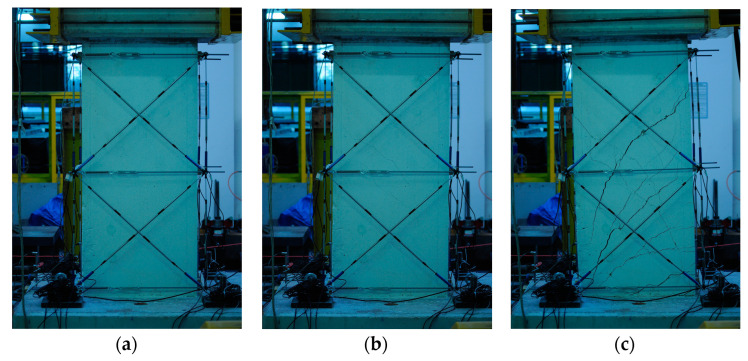
Pier images during the cyclic test at time: (**a**) test beginning; (**b**) first shear cracking at fourth drift and (**c**) final crack pattern—rectangular pier.

**Figure 19 materials-16-02790-f019:**
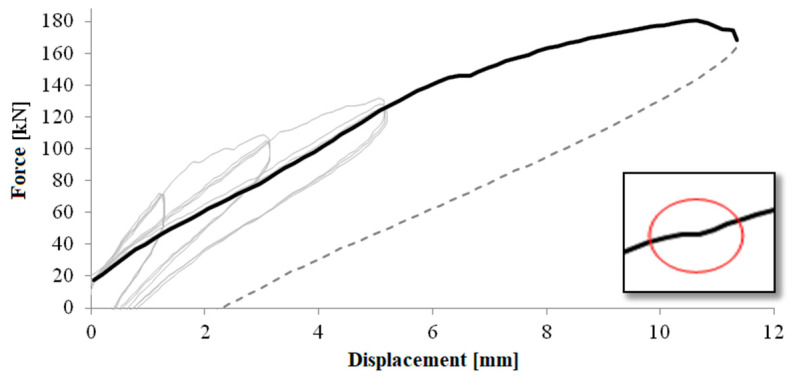
Load-displacement diagram at fourth drift (positive values)—rectangular pier.

**Figure 20 materials-16-02790-f020:**
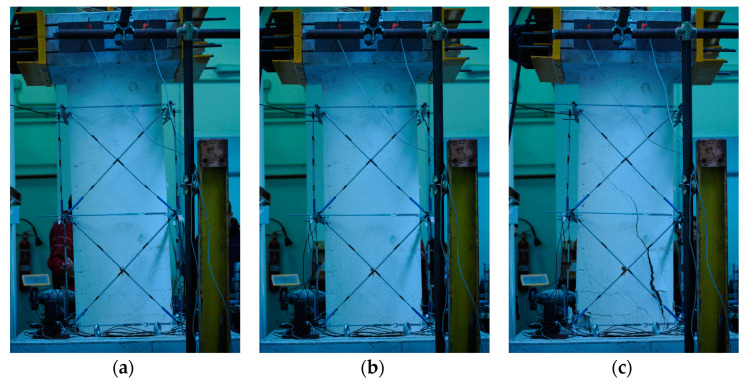
Pier images during the cyclic test at time: (**a**) test beginning, (**b**) first shear cracking at fourth drift and (**c**) final crack pattern—circular pier.

**Figure 21 materials-16-02790-f021:**
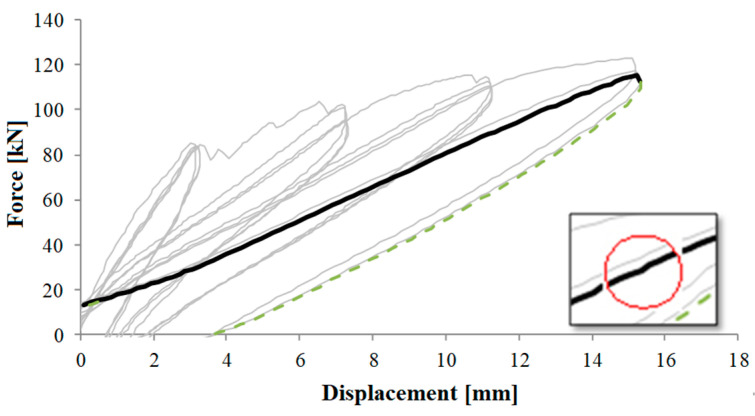
Load-displacement diagram at fourth drift (positive values)—circular pier.

**Figure 22 materials-16-02790-f022:**
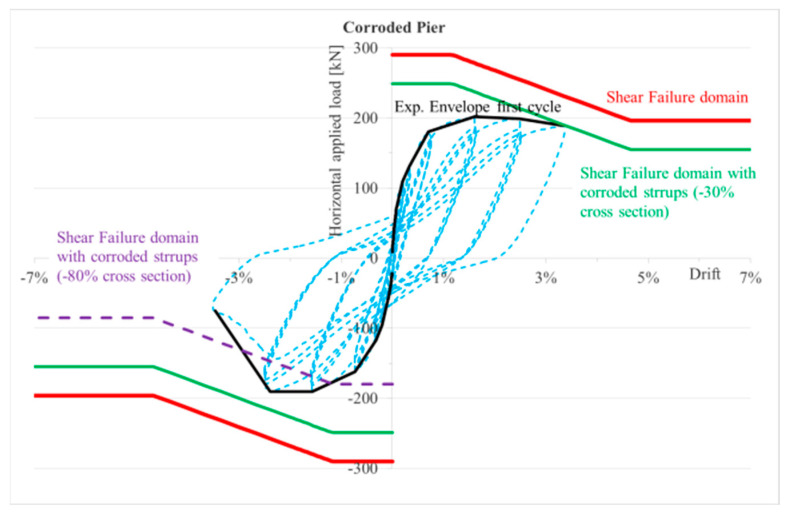
Envelope of first cycles at different drifts and shear failure domain at different cross-section reductions—rectangular pier.

**Table 1 materials-16-02790-t001:** Designed test protocol—rectangular pier.

Drift Level (%)	Displacement (mm)	Cycles No. (-)	Speed (mm/s)	Time (min)	Progressive Time (h)
0.15	2.25	3	0.50	0.90	0.02
0.30	4.50	3	0.50	1.80	0.05
0.45	6.75	3	0.50	2.70	0.09
0.90	13.50	3	1.00	2.70	0.14
1.80	27.00	3	1.00	5.40	0.23
2.70	40.50	3	1.00	8.10	0.36
3.60 *	54.00	3	1.00	10.80	0.54
4.50	67.50	3	1.00	13.50	0.77
5.40	81.00	3	1.00	16.20	1.04

* Test ended at 3.60% drift due to pier failure.

**Table 2 materials-16-02790-t002:** Designed test protocol—circular pier.

Drift Level (%)	Displacement (mm)	Cycles No. (-)	Speed (mm/s)	Time (min)	Progressive Time (h)
0.25	4.13	3	0.50	1.65	0.03
0.50	8.25	3	0.50	3.30	0.08
0.75	12.38	3	0.50	4.95	0.17
1.00	16.50	3	1.00	3.30	0.22
2.00	33.00	3	1.00	6.60	0.33
3.00	49.50	3	1.00	9.90	0.50
4.00 *	66.00	3	1.00	13.20	0.72
5.00	82.50	3	1.00	16.50	0.99

* Test ended at 4.00% drift due to pier failure.

**Table 3 materials-16-02790-t003:** Peak load at each cycle and drift level—rectangular pier.

Drift Level(%)	Drift 1 (0.15%)	Drift 2 (0.3%)	Drift 3 (0.45%)	Drift 4 (0.9%)	Drift 5 (1.8%)	Drift 6 (2.7%)	Drift 7 (3.6%)
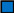 Cycle 1+	71.64	108.96	131.72	180.76	201.54	199.28	188.82
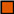 Cycle 1−	−59.12	−96.14	−117.02	−162.08	−190.56	−190.84	−75.06
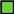 Cycle 2+	71.3	104.84	128.18	172.08	187.5	184.7	
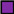 Cycle 2−	−58.72	−94.64	−113.28	−158.92	−182.46	−180	
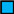 Cycle 3+	70.66	103.5	126.3	169.54	181.84	177.74	
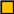 Cycle 3−	−58.5	−93.8	−111.66	−157.2	−179.04	−173.96	

**Table 4 materials-16-02790-t004:** Percentage variation, capacity loss—circular pier.

Between	Drift 1 (0.25%)	Drift 2 (0.50%)	Drift 3 (0.75%)	Drift 4 (1.0%)	Drift 5 (2.0%)	Drift 6 (3.0%)
Cycle 1+ and 2+	0.47%	3.78%	2.69%	4.80%	6.97%	7.32%
Cycle 1− and 2−	0.68%	1.56%	3.20%	1.95%	4.25%	5.68%
Cycle 2+ and 3+	0.90%	1.28%	1.47%	1.48%	3.02%	3.77%
Cycle 2− and 3−	0.37%	0.89%	1.43%	1.08%	1.87%	3.36%

## Data Availability

Data is contained within the article.

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
