# Peer review of "Reinforcement Corrosion in RC Hollow Piers: Destructive and Non-Destructive Tests"

_materials, 2023, doi:10.3390/ma16072790_

Round 1

Reviewer 1 Report

Journal: Materials MDPI

Title: Reinforcement corrosion in RC hollow piers: destructive and non-destructive tests.

The authors have presented the cyclic-load tests on reduced scale corroded reinforced concrete hollow cross- section bridge piers have been experimentally performed and compared to the results of similar non-corroded RC piers. The authors are attempting to illustrate the proper notion of how the aged and corroded piers fails due to cyclic loading.

However, it still needs some moderate clarification to make it publishable as scientific research. The concerns are presented as follows:

Key comments

1.      The introduction section seems insufficient information regarding the title. authors have explained the previous studies regarding the issues in short, despite mentioning so many previous studies…, Authors should clarify by evidence showing the gaping of previous studies for novelty.

2.      Because there is a lack of evidence regarding the investigation's novelty, it is suggested that it be made a little more elaborate.

3.      Chapter two: (Sections Geometry)- error in writing of sectional dimensions {2,40 m, 1,35 m} page 2, line 54,55,56 and 57). So, correct these as standard numerical values.

4.      Chapter 2, sections 2.2 and 2.3 are the same titles, so, it is better to combine them in one of 2.2 and 2.3. likewise; the composition and size of specimens and dimension format also suggested making them compactable and good visualizing

5.      The authors described the testing process and results of non-destructive test in the chapter three (3.1 and 3.2) with more explanation rather than compactable form. The presentation by figure and tables seems not proper formats (like figure 7, figure 9 looks not clear).  So, it is suggested to make it more clear with neat and clear diagram/ figures/ tables and only short narration.

6.      The authors have illustrated experimental and numerical analysis results with sufficient graphical and numerical values, but it does not look like a perfect composition of graphical presentation, equations (page 10. Eq. 2). That’s why it is suggested to make a short narration and link the result with numerical values and graphs, considering the purpose of the research. So, the reader can get it easily.

7.      It should be easier to understand the whole outcomes (experimental and theoretical data) if authors provide the comparison summary before conclusion chapter.

8.      In overall, it is strongly suggested to re-arrange the tables and figures with proper size with considering the visualization (color).

9.      The conclusion seems a little generic, so, the authors should try to present a more novel and concrete conclusion rather than a generic one.

10.  There are minor typos in the manuscript; checking the typos and grammatical odds thoroughly is suggested.

Reviewer 2 Report

1- abstract need to be rewritten. Add some numbers, increase or decrease.

2- literature review is very limited. Add state of the art review or previous studies.

3- add research contribution, originality and novelty. 

4- add actual photos or figs of specimens.

5- add coefficients of variance for all results reported.

6- conclusion is very general. Be specific to your studies.

Reviewer 3 Report

The submitted Article with the Manuscript ID: “Materials-2267398” and the title: “Reinforcement corrosion in RC hollow piers: destructive and non-destructive tests” investigates the degradation monitoring and the mechanical performance of reduced scale, corroded reinforced concrete hollow cross-section bridge piers under cyclic loads and results compared with similar non-corroded piers.

The paper has a good scientific approach to the subject. However, some issues, questions, and clarifications should be amended before publication, and it needs some improvement to reach the required scientific level. Nevertheless, as an overall comment based on the above remarks, the paper is worthy of publication after revision. Therefore, the following comments and suggestions are raised for the authors’ reference:

  1. More literature is needed in the introduction part. It is essential to state what has been done and what has not been done in the area of Structural Health Monitoring, focusing on damage investigation with Non-Destructive Evaluation techniques. Moreover, supplementary literature is needed utilizing the fracture characteristics of the microstructure of corroded concrete specimens after mechanical testing. Thus, the following additional studies are suggested to be considered (order by date):

- “Acoustic monitoring for the evaluation of concrete structures and materials,” Acoustic Emission and Related Non-Destructive Evaluation Techniques in the Fracture Mechanics of Concrete, 2015.

-  “Structural health monitoring of seismically vulnerable RC frames under lateral cyclic loading,” Earthquakes and Structures,  2020.

- “Research on in-situ corrosion process monitoring and evaluation of reinforced concrete via ultrasonic guided waves,” Construction and Building Materials, 2022

  1. In the 2.1 heading with the title ‘Sections Geometry,’ what exactly is the percentage % of the reduction of the scale factor used, and is the same for both the hollow piers, the Rectangular and the Circular, respectively?
  2. In Figure 2, the dimensions are missing.
  3. In Figure 2, a side photo or a schematic representation could help understand how the setup of the aging process has been applied to the piers because only one part of the stem of the two piers was subjected to accelerated corrosion.
  4. The Sonreb method should be described with the addition of some references.
  5. In line 93, an explanation of what is considered close to the corroded areas is needed. Since the distance between the controlled (healthy) areas from the accelerated ageing process area is of great importance, the exact distance between points 1A, 3A, and 1B, 2B, 3B should be mentioned and compared between the rectangular and the cyclical piers as well. They follow the same pattern of characterizing what is close-up and what are distant points for the two piers tested. A schematic representation with the exact distances between points of interest may be needed for both piers.
  6. A description of the techniques and instruments used for the ultrasound inspection and the methodology is needed.
  7. In Figure 6, why the UPV values of USP-1A and USP-3A are lower than USP-2B? An explanation is needed. Does the distance between 1A, 3A, and 2B from the corroded area affect the UPV?
  8. A schematic representation of the four points tested for the SONREB testing is also needed for the Circular hollow pier.
  9. In Figures 12 and 13, captions are needed to explain the different colors.
  10. In line 231, Figures 15, 16, and 17 show moment-curvature diagrams of the rectangular or cyclic pier accordingly?
  11. In Figures 20a and 20b, a better resolution is needed to distinguish the first crack and the crack pattern.
  12. How is the 0.68% reduction of longitudinal bar diameter due to corrosion been measured?

Reviewer 4 Report

This manuscript depicts the efforts made to understand the cyclic behaviour of RC piers with corroded steel reinforcement bars.

As a general comment, I must highlight the endeavour's originality and topicality and the results' usefulness. On the other hand, the manuscript needs significant improvements.

The following specific issues should also be taken into consideration:

-          The abstract should be written as a single paragraph;

-          "Infrastructures built before the 1980s were designed without taking into account the principles of seismic engineering and anti-seismic regulations or design codes". This sentence is not true. Seismic provisions in the past century were generally much simpler than current ones,  but there was very significant seismic design worldwide before 1980. Moreover, "anti-seismic" is inaccurate terminology;

-          Redaction is informal and should be enhanced to the level of a research article;

-          The introduction is exiguous and does not frame the theme accurately;

-          There is not a brief literature review or related work section that could support the research gap definition;

-          A critical discussion section is missing;

Reviewer 5 Report

  The article has a scientific character. The article deals with corrosion of pillar reinforcements. The Authors applied correct research methods and used the appropriate measuring equipment. The content of the work is logically written. The manuscript contains 24 figures and 4 tables. Figures and tables aren't properly prepared. Authors cited 22 literature sources. The authors presented an interesting work, but it requires numerous improvements to be of satisfactory quality. General remarks 1. The authors must clearly indicate the novelty of the work compared to the current state of the art 2. The research equipment used in the research should be properly described and characterized; Specify the data of the apparatus in the article in the following order: device designation, manufacturer's name, city, country. 3. In general, improve the discussion of the results obtained 4. The work also requires careful text editing 5. The quality of drawings should be improved at work: 4, 10, 18, 20, 21, 23 (remove photos of people from Fig. 21 and 23) 6. Conclusions in Conclusion should be given in the form of points 7. For Figures 6, 8, 12, 15, 16, 17, show the result points and specify the measurement uncertainty. 8. Fig. 5 - add markings of components 9. Figure 7, 9 add table row labels 10. Add the determination uncertainty for Tables 1, 2, 3 and 4 11. The work lacks a proper presentation of the state of the art; providing a collective list of literature on the subject without discussing them is inappropriate 12. The work lacks a proper analysis of reinforcement corrosion 13. Authors should show details of damage caused during testing. The authors must make corrections in accordance with the above comments and submit the work for re-evaluation.

Reviewer 6 Report

The paper investigates an interesting topic such as the reinforcement corrosion in RC hollow piers: destructive and 2 non-destructive tests. Howver, some issues need to be considered.

Abstract. No paragraphs are needed.

Introduction

This part needs to be extended with references. For example, the role of corrosion in the assessment of infrastructure resilience needs to be addressed.

The novelties of the paper needs to be discussed in order to support the originality of the study

Section 2.4

This sentence is not clear:

"The vertical load was applied by means of 3 D16 nominal high strength post-tensioned steel strands fixed to the base inside the foundation." please rewrite.

Figure 5 needs dimensions.

Section 3

Figures 8 and 9: the y axes need to be rescaled.

Section 5

Is Eqn.(1) proposed in the paper or does it come from a reference? In this case, please refer to it.

figure 19 the grey colours are difficult to be read.

Section 6

Is this section novel? If not, please shorten.

Also, consider to put this just after the introduction

Round 2

Reviewer 2 Report

Previous comments adressed

Author Response

Authors thank the reviewer for the positive comment

Reviewer 3 Report

The revised Article with the Manuscript ID: “Materials-2267398-v2” and the title: “Reinforcement corrosion in RC hollow piers: destructive and non-destructive tests has been improved substantially. The efforts performed by the authors to consider all the recommendations and to respond to all the criticisms of the previous review comments are greatly appreciated. The manuscript is well-structured, and it has adequate novelty presenting valuable results and helpful concluding remarks, and, therefore, the revised version of this study is worthy of publication. The paper is suggested to be accepted for publication in the journal without further re-review.

Author Response

(The authors gave the same response as above.)

Reviewer 4 Report

I thank the Authors for the improvement on the manuscript.

Those were less than what I expected, yet, I must recognize that the manuscript quality improved.

Author Response

(The authors gave the same response as above.)

Reviewer 5 Report

I thank the Authors for considering my comments. I will recommend the publication of the manuscript.

Author Response

(The authors gave the same response as above.)

Reviewer 6 Report

The authors answered to all my questions/obesrvations except for:

This part needs to be extended with references. For example, the role of corrosion in the assessment of infrastructure resilience needs to be addressed.

Please refer to:

Andisheh, Kaveh; Scott, Allan; Palermo, Alessandro Effects of Corrosion on Stress–Strain Behavior of Confined Concrete Journal of structural engineering. , 2021, Vol.147(7) ISSN: 0733-9445 , 1943-541X; DOI: 10.1061/(ASCE)ST.1943-541X.0003005

Forcellini, D. A new methodology to assess indirect losses in bridges subjected to multiple hazards. Innov. Infrastruct. Solut. 2019, 4, 1–9.

Matthews, Benjamin, Alessandro Palermo, and Allan Scott. “Overview of the Cyclic Response of Reinforced Concrete Members Subjected to Artificial Chloride‐induced Corrosion.” Structural concrete : journal of the FIB. 24.1 (2023): 100–114.

Author Response

Authors thank the reviewer for the suggestion; relevant references were added to the manuscript